# Exploring Vision Semantic Prompt for Efficient Point Cloud Understanding

Yixin Zha [1]  Chuxin Wang [1]  Wenfei Yang [1 2]  Tianzhu Zhang [1 2]  Feng Wu [1 2]

## Abstract

A series of pretrained models have demonstrated promising results in point cloud understanding tasks and are widely applied to downstream tasks through fine-tuning. However, full fine-tuning leads to the forgetting of pretrained knowledge and substantial storage costs on edge devices. To address these issues, Parameter-Efficient Transfer Learning (PETL) methods have been proposed. According to our analysis, we find that existing 3D PETL methods cannot adequately align with semantic relationships of features required by downstream tasks, resulting in suboptimal performance. To ensure parameter efficiency while introducing rich semantic cues, we propose a novel fine-tuning paradigm for 3D pretrained models. We utilize frozen 2D pretrained models to provide vision semantic prompts and design a new Hybrid Attention Adapter to efficiently fuse 2D semantic cues into 3D representations with minimal trainable parameters(1.8M). Extensive experiments conducted on datasets including ScanObjectNN, ModelNet40, and ShapeNetPart demonstrate the effectiveness of our proposed paradigm. In particular, our method achieves 95.6% accuracy on ModelNet40 and attains 90.09% performance on the most challenging classification split ScanObjectNN(PB-T50-RS).

## 1. Introduction

With the growing of training data and model parameters, large foundation models have achieved success across various domains and tasks. Point clouds, as direct representations of the real world, play a crucial role in various fields (Li et al., 2024; Pan et al., 2024). Inspired by pretrained models

---

[1]University of Science and Technology of China, Hefei, China [2]Deep Space Exploration Lab, Hefei, China. Correspondence to: Yixin Zha <zyxcn@mail.ustc.edu.cn>.

*Proceedings of the 42$^{nd}$ International Conference on Machine Learning*, Vancouver, Canada. PMLR 267, 2025. Copyright 2025 by the author(s).

in natural language processing (Devlin et al., 2018; Raffel et al., 2020; Achiam et al., 2023; Floridi & Chiriatti, 2020) and vision understanding (He et al., 2022; Radford et al., 2021; Oquab et al., 2023; Dehghani et al., 2023), similar methods of point cloud understanding have been proposed, such as PointBERT (Yu et al., 2022), PointMAE (Pang et al., 2022) and PointGPT (Chen et al., 2024). These works utilize large amounts of unlabeled data to learn general representations and apply them to downstream tasks through full fine-tuning. However, the full fine-tuning strategy faces two significant shortcomings: (1) Fine-tuning the entire model leads to the forgetting of pretrained knowledge; (2) Full fine-tuning imposes a significant storage burden.

To address the aforementioned constraints, a series of Parameter-Efficient Transfer Learning (PETL) (Hu et al., 2021; Jia et al., 2022; Chen et al., 2022) methods have been proposed. In point cloud understanding tasks, researchers have also proposed a series of PETL approaches, such as IDPT (Zha et al., 2023), DAPT (Zhou et al., 2024), and Point-PEFT (Tang et al., 2024). However, as shown in Figure 1(b), performance of the 3D PETL method on complex tasks remains unsatisfactory. Upon analysis, we believe this limitation may stem from the differences in feature requirements between point cloud pretraining tasks and downstream tasks. As shown in Figure 1(a), point cloud features of pretrained models are limited to local structural information and exhibit positional preferences. In contrast, features obtained through full fine-tuning contain rich semantic information, with consistent representations for components of the same structure. However, in PETL methods, the main backbones are frozen, resulting in output features with limited semantic cues, which constrains networks generalization ability in downstream tasks.

Compared to point clouds, 2D images contain richer semantic details. Currently, there are many existing 2D models pretrained on large-scale image datasets. These models not only learn rich semantic cues but are also widely deployed on edge devices. This raises an intriguing question worth exploring: *Can we leverage 2D semantics to enhance the performance of efficient 3D understanding?* We believe that effectively integrating 2D semantic cues with 3D features could significantly improve model performances, even with minimal trainable parameters. To explore the feasibility of

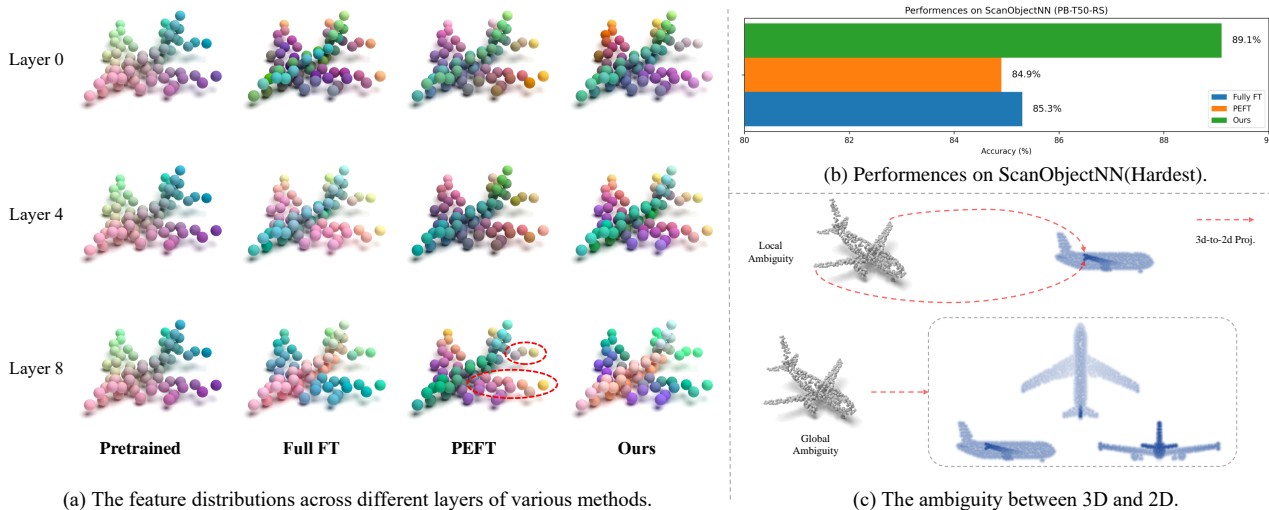

(b) Performances on ScanObjectNN(Hardest).

(a) The feature distributions across different layers of various methods.

(c) The ambiguity between 3D and 2D.

*Figure 1.* (a) The feature colors are transformed into feature space using PCA, where the same color indicates feature consistency. Features in red circles fail to maintain consistent. (b) IDPT (Zha et al., 2023) underperform Full FT approaches on the most challenging real-world classification tasks. (c) The inherent limitations of the 2D perspective during 3D-to-2D projection introduce ambiguities between local and global structures.

this design, we conduct an analysis of technical challenges:

- **Ambiguities in 3D-to-2D projection:** The 2D projections of point clouds from different viewpoints introduce local and global ambiguities. As shown in Figure 1(c), two points that are far apart in 3D space may appear very close to each other on 2D planes from certain viewpoints, which causes local ambiguities. Additionally, viewpoint variations induce changes in the projected 2D geometry of 3D objects, resulting in global ambiguities due to inconsistent representations across perspectives.

- **Multimodal Fusion in PETL:** Traditional multimodal approaches typically rely on learnable modality-specific feature extractors to achieve feature fusion across modalities. However, in PETL, most parameters of the feature extractors are frozen. Achieving effective multimodal feature fusion with minimal tunable parameters presents a significant challenge.

Based on our above analysis, we propose a novel paradigm that leverages visual semantic prompts to improve the generalization of pretrained 3D models while keeping parameter efficiency. The new paradigm includes three new designs: 3D-to-2D Projection, Vision Semantic Prompt and Hybrid Attention Adapter (HAA). We first map point clouds into 2D depth maps from three orthogonal viewpoints to mitigate global ambiguities. Then, both point clouds and their corresponding depth maps are fed into the network. We adopt a multi-scale semantic cues injection strategy, as shown in Figure 3, each layer consists of two parallel transformers with different modality weights, each transformer handles infor-

mation from one of two modalities. On each scale, vision semantic prompts are generated by a non-linear layer with 2D class tokens, and we employ HAA to achieve modality fusion. In HAA, prompts are passed through non-linear layers to generate two learnable parameters, $\alpha$ and $\beta$. 3D features are modulated by these parameters to achieve Semantic Transfer (ST), which decouples semantic cues from global contextual features to mitigate local ambiguities. The modulated features are then used as queries and keys to compute the self-similarity. The unaltered 3D features serve as values, which are updated using the similarity matrixes acquired above. This approach enhances the semantic associations of 3D features while effectively filtering out redundant 2D noise, and the trainable parameters are kept at an extremely low level (1.8M).

In summary, our main contributions are as follows: (1) We propose a new paradigm that, for the first time, leverages 2D semantic cues to improve the generalization of pretrained 3D models with minimal trainable parameters. (2) We utilize 2D class tokens at multiple scales to generate prompts, and we design a Hybrid Attention Adapter to adopt efficient modality fusion while keeping the trainable parameters at an extremely low level. (3) Extensive experiments on datasets such as ScanObjectNN, ModelNet40, and ShapeNetPart demonstrate the effectiveness of our proposed paradigm.

## 2. Related Work

**Large-scale Pretrained Models:** Large-scale pretrained models have demonstrated exceptional performance on downstream tasks across various domains, including natural language processing (NLP) (Devlin et al., 2018; Raffel

et al., 2020; Achiam et al., 2023; Floridi & Chiriatti, 2020), 2D vision (Radford et al., 2021) (Dehghani et al., 2023), and point cloud understanding (Zhang et al., 2022). DI-NOv2 (Oquab et al., 2023) employed visual transformers to perform self-supervised pre-training, while MAE (He et al., 2022) achieved pre-training by randomly masking parts of an image and reconstructing the masked pixels using the remaining visible portions. Recently, many self-supervised pre-training methods for point clouds have been proposed. Existing methods primarily follow two research paths, contrastive learning method (Xie et al., 2020; Zhang et al., 2021; Dong et al., 2023) and generative methods (Yu et al., 2022; Pang et al., 2022; Zhang et al., 2023b). Contrastive learning methods guide the model in learning discriminative features by distinguishing positive and negative samples. For example, PointContrast (Xie et al., 2020) constrains the consistency between the same points in different views. CrossNet (Wu et al., 2023) conducts cross-modal contrastive learning between point clouds and their corresponding rendered images. Motivated by BERT (Devlin et al., 2018) and MAE (He et al., 2022), generative methods mainly adopt the Masked Point Modeling (MPM) to encourage models to infer the randomly masked regions with the visible regions, thereby guiding the model to learn the relationships between point cloud patches in the process, such as Point-MAE (Pang et al., 2022), PointMamba (Liang et al., 2024) and PointBERT (Yu et al., 2022) However, these point cloud pretrained models may forget pre-training knowledge after full fine-tuning, and impose a significant storage burden.

**Parameter-effective Transfer Learning:** The pre-training and fine-tuning paradigm has demonstrated remarkable effectiveness across a wide range of tasks. However, as model sizes grow exponentially, full fine-tuning the entire model can cause significant storage burdens. In contrast, Parameter-Efficient Transfer Learning (PETL) methods (Hu et al., 2021; Jia et al., 2022; Chen et al., 2022; Liu et al., 2023; Houlsby et al., 2019) update only a small subset of the model's parameters while keeping the rest frozen. These approaches have demonstrated both effectiveness and efficiency across various widely-used pretrained models, including BERT (Devlin et al., 2018), GPT series (Achiam et al., 2023; Floridi & Chiriatti, 2020), ViT (Dosovitskiy et al., 2020), CLIP (Radford et al., 2021), and Stable Diffusion (Rombach et al., 2022). PETL methods can typically be divided into three main categories: prompt tuning (Jia et al., 2022; Yang et al., 2024), reparameterization (Hu et al., 2021), and adapters (Chen et al., 2022; Zhang et al., 2023a). These techniques adapt pretrained models to specific tasks by fine-tuning prompts, adjusting parameters without changing the model architecture, or inserting lightweight trainable layers, respectively. Recently, PETL techniques have been introduced into the 3D domain, such as IDPT (Zha et al., 2023), DAPT (Zhou et al., 2024) and Point-PEFT (Tang et al., 2024). However, in 3D PETL methods, the backbones are frozen, resulting in output features with limited semantic cues, which constrains the network's generalization ability in downstream tasks. We believe that the rich semantic cues in 2D pretrained models can effectively compensate for the missing semantic information in 3D PETL, and this area of exploration remains untapped.

## 3. Method

We first introduce the transformer-based paradigms of 3D pretrained models in Sec. 3.1. Next, we discuss the paradigms of fine-tuning in Sec. 3.2, including Parameter-Efficient Transfer Learning. Then we elaborate on the process of projecting 3D point clouds onto 2D planes in Sec. 3.3. Finally, we delve into the details of the multi-scale modality fusion framework in Sec. 3.4, including Tokenizer, Vision Semantic Prompt Generation and Hybrid Attention Adapter.

### 3.1. Transformer-based 3D Pretrained Model

In pretrained transformer-based point cloud models, a point cloud $P \in \mathcal{R}^{N \times 3}$ with $N$ points is first divided into $n$ point patches $p \in \mathcal{R}^{n \times k \times 3}$ via Farthest Point Sampling (FPS) and K-Nearest Neighborhood (KNN) algorithms, where each patch contains $k$ local points. Then, all point patches will be embedded into a token sequence $T_{3D} \in \mathcal{R}^{n \times C}$ through mini-PointNet (Qi et al., 2017). The sequence is further processed by L-layer transformer blocks. After that, point tokens are updated through attention layers. Outputs of attention layers are passed through a FeedForward Network (FFN) with residual connections to extract channel-wise information. The transformer block can be written as:

$$\hat{T}_i = \text{Attention}(\text{LN}(T_{i-1}) + T_{i-1}),$$
$$T_i = \text{FFN}(\text{LN}(\hat{T}_i)) + \hat{T}_i, \tag{1}$$

where $T_i$ is the output of $i$-th transformer block, $LN$ is a Layer Normalization layer.

### 3.2. Fine-Tuning Paradigms

**Full Fine-tuning:** Full fine-tuning is the most commonly used fine-tuning paradigm. Assuming the training setup for the downstream task is configured as $\Gamma(x; y)$ (i.e., $x$ are training data and $y$ are labels) and pretrained model weights as $\theta$. After fine-tuning, all trainable parameters $\theta$ of pretrained models are updated to $\hat{\theta}$. Specifically, the full fine-tuning paradigm can be formulated as:

$$\hat{\theta} = \arg \min_{\theta} \ell\big(F(x; \theta), y\big), \tag{2}$$

where $\ell$ represents the loss function for downstream tasks. Each fine-tuning iteration requires storing the entire model's parameters. As the model size increases, this results in significant storage consumption.

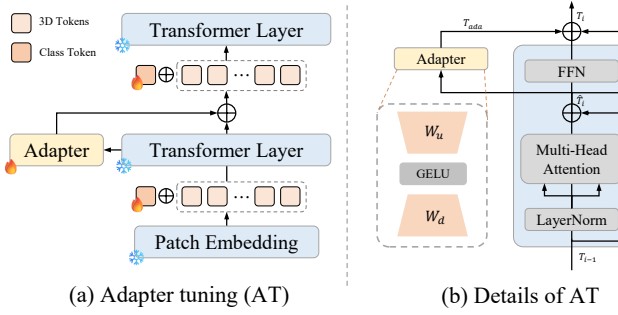

(a) Adapter tuning (AT)    (b) Details of AT

*Figure 2.* The illustration of Adapter Tuning: (a) The overview of Adapter for trnasformer-based architecture. (b) The details of transformer block with Adapter.

**Parameter-Efficient Transfer Learning:** PETL methods offer an efficient approach to adapting pretrained models for downstream tasks. Existing PETL methods focus on tuning only a tiny subset of model weights or some lightweight additional parameters, we denote these tunable parameters as $\theta^*$. The optimized parameters $\hat{\theta}^*$ can be represented as:

$$\hat{\theta}^* = \arg\min_{\theta^*} \ell\big(F(x;\theta,\theta^*),y\big), |\theta^*| << |\theta|. \quad (3)$$

The shape of $\theta^*$ is significantly smaller than pretrained model weights $\theta$. During fine-tuning, the pretrained parameters $\theta$ remain fixed while only $\theta^*$ are updated to $\hat{\theta}^*$.

There are two common PETL paradigms: Adapter Tuning (AT) (He et al., 2021; Zhang et al., 2023a; Chen et al., 2022) and Prompt Tuning (PT) (Jia et al., 2022; Yang et al., 2024). Here, we briefly introduce the AT paradigm. As illustrated in Figure 2(a), AT incorporates a small number of parameters into the transformer architecture by introducing a lightweight bottleneck module. Specifically, as shown in Figure 2(b), it consists of a downward projection $W_d$ to reduce the feature dimension, a non-linear activation function $\phi(\cdot)$, and an upward projection $W_u$ to restore the features to their original dimension. During fine-tuning, the $T_{3D}$ are concatenated with a learnable class token and form $T_{input} \in \mathcal{R}^{(n+1)\times C}$. Specifically, on the $i$-th layer, given input tokens $T_{i-1} \in \mathcal{R}^{(n+1)\times C}$, the calculation process inside Adapter can be formulated as:

$$T_{ada} = (W_u(\phi(W_d\hat{T}_i^T)))^T. \quad (4)$$

where the $\hat{T}_i \in \mathcal{R}^{(n+1)\times C}$ is the output of the attention module, and the $T_{ada}$ is the output of the adapter. Denote the $W_d \in \mathcal{R}^{d\times C}$ and the $W_u \in \mathcal{R}^{C\times d}$, the $d << C$. Besides, $\hat{T}_i \in \mathcal{R}^{(n+1)\times C}$ are fed into the FFN layer, and the output of the FFN are summed with $T_{ada}$ and $\hat{T}_i$, forming the final output of the transformer block $T_i$.

### 3.3. 3D-to-2D Projection

To enable 2D pretrained models to capture the semantic information of point clouds, we first map the point cloud into 2D depth maps. Given a point cloud $P$ and a cam-

era pose $V$, we aim to generate a 2D depth map $D^V(P)$ whose pixels $P$'s geometry that is visible in $V$. With the extrinsic and intrinsic parameters of the pose $V$, we can obtain a projective relationship between each 3D point and its corresponding 2D coordinate(i.e., deitals in B.2). Each 3D point $p(u, v, z) \in P$ is projected onto a projection plane, retrieving a 2D pixel location $(\hat{u}, \hat{v})$ and a depth $\hat{z}$ (a.k.a. distance from the projection). Projected 2D points $\hat{p}(\hat{u}, \hat{v})$ with depth $\hat{z}$ are used to generate a rendered image. Let $(x, y)$ be the coordinate of the rendered pixel, the generation process of whole 2D images $D^V$ can be formulated as:

$$D_{x,y}^V(P) = \max_{p \in P}\{||(x,y),\hat{p}||_2 \times \hat{z}_{neg}, 0\}.$$
$$\hat{z}_{neg} = 1 - (\hat{z} - \min_{p \in P}\hat{z})/(\max_{p \in P}\hat{z} - \min_{p \in P}\hat{z}), \quad (5)$$

The $\hat{z}_{neg}$ is a negative point depth normalized within [0, 1]. In our model, to overcome the ambiguity caused by viewpoint limitations, we project point clouds into 2D depth maps from three orthogonal viewpoints and feed all of them into the network with corresponding point clouds.

### 3.4. Multi-Scale Modality Fusion

In this section, we provide a detailed introduction to our proposed paradigm, which, for the first time, achieves enhanced 3D model performance by integrating 2D semantic cues with minimal trainable parameters.

**Tokenizer:** Given point clouds $P$, after 3D-to-2D projection, we can acquire depth projections of $P$ from three orthogonal viewpoints. Point clouds and their corresponding depth maps are fed into the model simultaneously. The point cloud is processed into tokens $T_{3D} \in \mathcal{R}^{n\times C}$ through the patch embedding of the 3D pretrained model, while three depth maps are separately processed by the 2D pretrained model into $r$ patch tokens follow the patch embedding of ViT (Dosovitskiy et al., 2020) and form $I \in \mathcal{R}^{3\times r\times D}$. The $T_{3D}$ and $I$ are each concatenated with learnable class tokens. These combined tokens $T_{input} \in \mathcal{R}^{(n+1)\times C}$ and $I_{input} \in \mathcal{R}^{3\times (r+1)\times D}$ are then updated layer by layer using their modality-specific transformers.

**Vision Semantic Prompt Generation:** During fine-tuning, each of the three depth maps independently performs self-attention computations, generating three separate 2D class tokens $I_{cls} \in \mathcal{R}^{3\times D}$. This operation reduces computational overhead and mitigates semantic ambiguities caused by viewpoint variations.

Taking one of the model layers as an example, after obtaining the updated 2D class tokens $I_{cls}$ with the depth maps, we first apply max-pooling on three tokens to select a single 2D class token $i_{cls} \in \mathcal{R}^{1\times D}$, The $i_{cls}$ is then passed through a shared Multi-Layer Perceptron (MLP) across all layers to map the high-dimensional 2D semantic features into the 3D semantic feature space. The whole process can

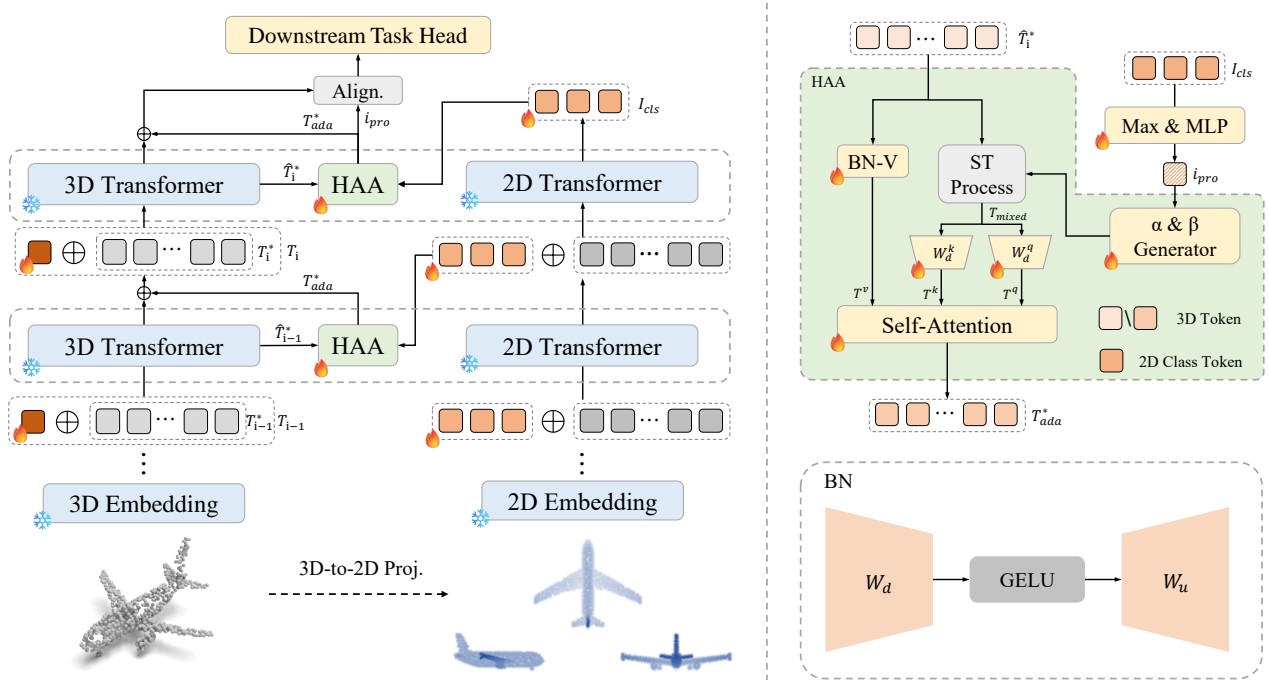

*Figure 3.* **Pipeline of our proposed framework.** Each layer consists of two parallel transformers, each transformer handles information from one of two modalities. For $i$-th layer, we employ HAA to achieve modality fusion. The $I_{cls}$ are processed through max pooling to obtain $i_{cls}$, and $i_{cls}$ are updated by an MLP to acquire vision semantic prompts $i_{pro}$. Then, $i_{pro}$ are passed through two parallel non-linear layers to generate two learnable parameters, $\alpha$ and $\beta$. These parameters are fed into HAA and modulate 3D tokens $\hat{T}_i^*$ to acquire $T_{mixed}$. $T_{mixed}$ are then used as queries and keys to compute the self-similarity. The unaltered $\hat{T}_i^*$ serve as values, which are updated using the calculated similarity matrix. We use the $i_{pro}$ of the last scale to perform the classification task to align modality semantics.

be formulated as:

$$i_{pro} = \text{MLP}(\max(I_{cls})). \qquad (6)$$

The updated 2D semantic prompt $i_{pro} \in \mathcal{R}^{1 \times C}$ is then fed into the Hybrid Attention Adapter within the same layer.

**Hybrid Attention Adapter:** Inspired by Adapter Tuning, we propose Hybrid Attention Adapter (HAA) to achieve efficient modality fusion. After obtaining 2D semantic prompts from frozen 2D pretrained models, both $i_{pro}$ and part of $\hat{T}_i$, the $\hat{T}_i^* \in \mathcal{R}^{n \times C}$ (i.e., $\hat{T}_i$ without class token) are fed into HAA simultaneously. Inside the HAA, the $i_{pro}$ undergoes two parallel non-linear layers to generate two learnable parameters, $\alpha$ and $\beta$. These parameters transfer 2D visual semantic cues to the 3D modal by modulating the normalized 3D features and obtain $T_{mixed}$. This operation decouples semantic information from global contextual features, mitigates the inherent local ambiguities in 3D-to-2D projection. The Semantic Transfer (ST) process can be formulated as:

$$T_{mixed} = \alpha * \left(\frac{\hat{T}_i^* - \mu(\hat{T}_i^*)}{\sigma(\hat{T}_i^*)}\right) + \beta, \qquad (7)$$

where $\mu(\cdot)$ and $\sigma(\cdot)$ are mean and standard deviation. Subsequently, we perform hybrid attention computations to achieve efficient modality fusion. We use the semantically

enriched 3D features $T_{mixed}$ as the query and key, and the unaltered $\hat{T}_i^*$ as the value, The novel attention mechanism can be formulated as:

$$T_{ada}^* = \text{Softmax}\left(\frac{T^q (T^k)^T}{\sqrt{s}}\right) T^v,$$
$$T^q = (W_d^q T_{mixed}^T)^T,$$
$$T^k = (W_d^k T_{mixed}^T)^T, \qquad (8)$$
$$T^v = (W_u^v (\phi(W_d^v \hat{T}_i^{*T})))^T,$$

where $s$ is the attention scale. This process updates the semantic relationships of the 3D features while filtering out redundant noise from the 2D modal. Inspired by Formula 4, we replace linear layers in transformer blocks with BottleNeck (BN) modules, this design allows us to conveniently balance the size of trainable parameters and model performance.

**Modality Semantic Alignment:** In the final layer of the model, 2D semantic prompts $i_{pro}$ are combined with 3D features. The combination are then fed into the downstream head to achieve alignment between two modality features. In classification tasks, $i_{pro}$ are concatenated with input of task head. In the part segmentation task, 2D prompts are used to perform classification tasks independently without

*Table 1.* Classification on three variants of the ScanObjectNN (Uy et al., 2019) and the ModelNet40 (Wu et al., 2015), including the number of trainable parameters, FLOPs and overall accuracy (OA). ALL methods utilize the default data argumentation as the baseline. **Red** indicates the best performance among all methods, while **Black** denotes the highest performance within PETL methods. D represents DINOv2 and C represents CLIP. Methods with $^\dagger$ using rotation data augmentation on ScanObjectNN.

| Method | Reference | Tunable params. (M) | ScanObjectNN | | | ModelNet40 | |
| --- | --- | --- | --- | --- | --- | --- | --- |
| | | | OBJ_BG | OBJ_ONLY | PB_T50_RS | Points Num. | OA (%) |
| *Supervised Learning Only* | | | | | | | |
| PointNet | CVPR 17 | 3.5 | 73.3 | 79.2 | 68.0 | 1k | - / 89.2 |
| PointNet++ | NeurIPS 17 | 1.5 | 82.3 | 84.3 | 77.9 | 1k | - / 90.7 |
| DGCNN | TOG 19 | 1.8 | 82.8 | 86.2 | 78.1 | 1k | - / 92.9 |
| MVTN | ICCV 21 | 11.2 | - | - | 82.8 | 1k | - / 93.8 |
| PointNeXt | NeurIPS 22 | 1.4 | - | - | 87.7 | 1k | - / 94.0 |
| PointMLP | ICLR 22 | 13.2 | - | - | 85.4 | 1k | - / 94.5 |
| RepSurf-U | CVPR 22 | 1.5 | - | - | 84.3 | 1k | - / 94.4 |
| ADS | ICCV 23 | - | - | - | 87.5 | 1k | - / 95.1 |
| *Self-Supervised Representation Learning (Full fine-tuning)* | | | | | | | |
| OcCo | ICCV 21 | 22.1 | 84.85 | 85.54 | 78.79 | 1k | - / 92.1 |
| Point-BERT | CVPR 22 | 22.1 | 87.43 | 88.12 | 83.07 | 1k | - / 93.2 |
| MaskPoint | ECCV 22 | 22.1 | 89.70 | 89.30 | 84.60 | 1k | - / 93.8 |
| Point-MAE | ECCV 22 | 22.1 | 90.02 | 88.29 | 85.18 | 1k | - / 93.8 |
| Point-M2AE | NeurIPS 22 | 15.3 | 91.22 | 88.81 | 86.43 | 1k | - / 94.0 |
| ACT$^\dagger$ | ICLR 23 | 22.1 | 93.29 | 91.91 | 88.21 | 1k | - / 93.7 |
| RECon$^\dagger$ | ICML 23 | 43.6 | 94.15 | 93.12 | 89.73 | 1k | - / 93.9 |
| PointMamba$^\dagger$ | NeurIPS 24 | 12.3 | 94.32 | 92.60 | 89.31 | 1k | 93.6 / - |
| *Self-Supervised Representation Learning (Parameter-Efficient Transfer Learning)* | | | | | | | |
| Point-BERT (baseline) | CVPR 22 | 22.1 | 87.43 | 88.12 | 83.69 | 1k | 92.7 / 93.2 |
| + IDPT | ICCV 23 | 1.7 (7.69%) | 88.12 | 88.30 | 83.69 | 1k | 92.6 / 93.4 |
| + Point-PEFT | AAAI 24 | 0.6 (2.71%) | - | - | 85.00 | 1k | 93.4 / - |
| + DAPT | CVPR 24 | 1.1 (4.97%) | 91.05 | 89.67 | 85.43 | 1k | 93.1 / 93.6 |
| + Ours(C) | - | 1.8 (1.04%) | **92.08** | 90.83 | 89.03 | 1k | 94.7 / 95.2 |
| + Ours(D) | - | 1.8 (1.66%) | 91.88 | 90.85 | 88.79 | 1k | 94.2 / 94.7 |
| Point-MAE (baseline) | ECCV 22 | 22.1 | 90.02 | 88.29 | 85.18 | 1k | 93.2 / 93.8 |
| + IDPT | ICCV 23 | 1.7 (7.69%) | 91.22 | 90.02 | 84.94 | 1k | 93.3 / 94.4 |
| + Point-PEFT | AAAI 24 | 0.6 (2.71%) | - | - | 85.50 | 1k | 94.2 / - |
| + DAPT | CVPR 24 | 1.1 (4.97%) | 90.88 | 90.19 | 85.08 | 1k | 93.5 / 94.0 |
| + Ours(C) | - | 1.8 (1.04%) | 91.86 | **91.20** | **89.14** | 1k | 95.2 / 95.6 |
| + Ours(D) | - | 1.8 (1.66%) | 91.95 | 90.89 | 89.07 | 1k | 94.6 / 95.2 |
| + Ours(C)$^\dagger$ | - | 1.8 (1.04%) | 92.33 | 91.83 | 90.09 | - | - |

affecting segmentation pipeline (i.e., details in B.1).

## 4. Experiments

In this section, we first present the implementation details in Sec. 4.1. After that, in Sec. 4.2, to demonstrate the effectiveness of the proposed paradigm, we evaluate its performance using four combinations of 2D and 3D pre-trained models on four downstream tasks, including synthetic object classification, real-world object classification, part segmentation and few-shot learning. We also carry on ablation studies for

the proposed paradigm in Sec. 4.3 to verify the effectiveness of proposed modules.

### 4.1. Implementation Details

For a fair comparison, all baselines adopt the same experimental setting: Freezing the pretrained 2D and 3D backbones while only updating identical newly inserted adapters and position embedding layer of the 2D pretrained model. We select two 2D and two 3D pretrained models respectively, and conduct four sets of experiments on each down-

*Table 2.* Part segmentation on the ShapeNetPart (Yi et al., 2016). The mIoU for all classed (Cls.) and for all instances (Inst.) are reported. #TP represents the tunable parameters. **Red** indicates the best performance among all methods, while **Black** denotes the highest performance within PETL methods. D represents DINOv2 and C represents CLIP.

| Method | Reference | #TP (M) | Cls.mIoU (%) | Inst.mIoU (%) |
|---|---|---|---|---|
| *Supervised Learning Only* | | | | |
| PointNet | CVPR 17 | - | 80.39 | 83.7 |
| PointNet++ | NeurIPS 17 | - | 81.85 | 85.1 |
| DGCNN | TOG 19 | - | 82.33 | 85.2 |
| APES | CVPR 23 | - | 83.67 | 85.8 |
| *Self-Supervised Representation Learning (Full fine-tuning)* | | | | |
| OcCo | ICCV 21 | 27.06 | 83.42 | 85.1 |
| Point-BERT | CVPR 22 | 27.06 | 84.11 | 85.6 |
| MaskPoint | ECCV 22 | - | 84.60 | 86.0 |
| Point-MAE | ECCV 22 | 27.06 | 84.19 | 86.1 |
| ACT | ICLR 23 | 27.06 | 84.66 | 86.1 |
| PointMamba | NeurIPS 24 | - | 84.40 | 86.2 |
| *Self-Supervised Representation Learning (Parameter-Efficient Transfer Learning)* | | | | |
| Point-BERT (baseline) | CVPR 22 | 27.06 | 84.11 | 85.6 |
| + IDPT | ICCV 23 | 5.69 | 83.50 | 85.3 |
| + DAPT | CVPR 24 | 5.65 | 83.83 | 85.5 |
| + Ours(C) | - | 6.65 | 84.61 | **86.2** |
| + Ours(D) | - | 6.65 | 84.52 | 86.1 |
| Point-MAE (baseline) | ECCV 22 | 27.06 | 84.19 | 86.1 |
| + IDPT | ICCV 23 | 5.69 | 83.79 | 85.7 |
| + DAPT | CVPR 24 | 5.65 | 84.01 | 85.7 |
| + Ours(C) | - | 6.65 | **84.70** | **86.3** |
| + Ours(D) | - | 6.65 | **84.73** | **86.2** |

stream task to validate the generalizability of the proposed paradigm. For the 3D pretrained models, we chose Point-MAE (Jiang et al., 2023) and PointBERT (Yu et al., 2022). For the 2D pretrained models, we select the CLIP (Radford et al., 2021) image encoder and DINOv2 (Oquab et al., 2023), where we use the ViT-B/16 version for CLIP and the ViT-B/14 version for DINOv2. All experiments are conducted on a single GeForce TRX 3090.

## 4.2. Effectiveness on Downstream Tasks

For all tasks, we report the results of four combinations: PointMAE + CLIP, PointMAE + DINOv2, PointBERT + CLIP and PointBERT + DINOv2.

**Real-World Shape Classification:** ScanObjectNN (Uy et al., 2019)is one of the most challenging 3D datasets, which covers 15K real-world objects from 15 categories. We report classification results of three variants. As Shown in Table 1, with comparable trainable parameters, the perfomance of proposed paradigm boost performances of 3D pretrained models on real-world shape classification tasks. It is worth noting that the combination "PointMAE + CLIP" achieves a performance of 89.14% on the most challenging split (PB-T50-RS), representing a 3.64% improvement over the previous state-of-the-art method Point-PEFT (85.5%).

Besides, if we adopt the same data augmentation strategy with *Recon* and *ACT*, the performance of "PointMAE + CLIP" surpasses all results. This demonstrates that the introduced vision semantic cues significantly enhance the generalization of 3D pretrained models on downstream tasks.

**Synthetic Shape Classification:** In addition to the experiments conducted on a real-world dataset, we perform experiments on a synthetic dataset, ModelNet40 (Wu et al., 2015), which consists of 12,311 clean 3D CAD models, covering 40 object categories. For testing the fine-tuned model, we provide results with and without the voting trick (Liu et al., 2019). The voting trick involves sampling multiple point clouds for the same sample and making model predictions multiple times, then aggregating the predictions through voting to obtain the final classification result. As Shown in Table 1, it can be observed that the proposed paradigm effectively improve the perfomance of pretrained across different combinations. The combination "PointMAE + CLIP" achieves a performance of 95.2%/95.6%, which is the highest perfomance across all full fine-tuning and supervised methods.

**Part Segmentation:** We conduct part segmentation experiments on the challenging ShapeNetPart (Yi et al., 2016) dataset, which comprises 16880 models with 16 different shape categories and 50 part labels. Experimental results on the ShapeNetPart dataset are shown in Table 2. The proposed paradigm boost the performance of 3D pretrained on the dataset, which is one of the hardest task. The "Point-MAE + CLIP" combination outperforms all methods in various experimental settings, demonstrating that vision semantic prompts can effectively introducing part semantics.

**Few-shot Classification:** To evaluate the effectiveness of the proposed modules with limited finetuning data, we conduct experiments for few-shot classification on ModelNet40. As shown in Table 3, the proposed paradigm boost the performance of models on 10-way k-shot settings, and achieve comparable result on 5-way k-shot. The results illustrate that our approach can augment pretrain models generalization capabilities by introducing semantic cues.

## 4.3. Ablation Study

As shown in Table 4, we systematically evaluate the contribution of each component in our proposed framework through four-phase ablation studies. *(1) 2D Baseline:* To evaluate 2D model capabilities, we exclusively utilize the CLIP image encoder for classification, with inputs being three orthogonal depth maps projected from point clouds. *(2) + 3D Model:* Building upon phase 1, we concatenate features from both the frozen Point-MAE encoder with DAPT (Zhou et al., 2024) and CLIP encoder before feeding them to the classification head. *(3) + Semantic Transfer:* To evaluate the influence of vision semantic prompts, we use

*Table 3.* Few-shot learning on ModelNet40 (Wu et al., 2015). We report overall accuracy (%) ± the standard deviation (%) over ten runs. D represents DINOv2 and C represents CLIP.

| Method | Reference | 5-way | | 10-way | |
|---|---|---|---|---|---|
| | | 10-shot | 20-shot | 10-shot | 20-shot |
| *Self-Supervised Representation Learning (Full fine-tuning)* | | | | | |
| OcCo | ICCV 21 | 94.0±3.6 | 95.9±2.3 | 89.4±5.1 | 92.4±4.6 |
| Point-BERT | CVPR 22 | 94.6±3.1 | 96.3±2.7 | 91.0±5.4 | 92.7±5.1 |
| MaskPoint | ECCV 22 | 95.0±3.7 | 97.2±1.7 | 91.4±4.0 | 93.4±3.5 |
| Point-MAE | ECCV 22 | 96.3±2.5 | 97.8±1.8 | 92.6±4.1 | 95.0±3.0 |
| Point-M2AE | NeurIPS 22 | 96.8±1.8 | 98.3±1.4 | 92.3±4.5 | 95.0±3.0 |
| ACT | ICLR 23 | 96.8±2.3 | 98.0±1.4 | 93.3±4.0 | 95.6±2.8 |
| RECon | ICML 23 | 97.3±1.9 | 98.9±3.9 | 93.3±3.9 | 95.8±3.0 |
| *Self-Supervised Representation Learning (Parameter-Efficient Transfer Learning)* | | | | | |
| Point-BERT (baseline) | CVPR 22 | 94.6±3.1 | 96.3±2.7 | 91.0±5.4 | 92.7±5.1 |
| + IDPT | ICCV 23 | 96.0±1.7 | 97.2±2.6 | 91.9±4.4 | 93.6±3.5 |
| + DAPT | CVPR 24 | 95.8±2.1 | 97.3±1.3 | 92.2±4.3 | 94.2±**3.4** |
| + Ours(C) | - | **96.3**±3.2 | **97.5**±2.5 | **93.1**±4.2 | 95.0±4.8 |
| + Ours(D) | - | 96.1±3.5 | 97.1±3.2 | 93.0±5.5 | **95.2**±**3.8** |
| Point-MAE (baseline) | ECCV 22 | 96.3±2.5 | 97.8±1.8 | 92.6±4.1 | 95.0±3.0 |
| + IDPT | ICCV 23 | **97.3**±**2.1** | 97.9±1.1 | 92.8±4.1 | 95.4±2.9 |
| + DAPT | CVPR 24 | 96.8±**1.8** | 98.0±1.0 | 93.0±3.5 | 95.5±3.2 |
| + Ours(C) | - | 97.0±3.2 | **98.3**±1.8 | **93.8**±**4.0** | **96.8**±**3.2** |
| + Ours(D) | - | 96.8±2.5 | 98.0±2.0 | 93.6±3.8 | 96.5±3.0 |

*Table 4.* Ablation learning of the proposed paradigm. The detail of the table are described in section 4.3.

| Method | #TP (M) | PB_T50_RS |
|---|---|---|
| 3D Baseline | 1.07 | 85.18 |
| + 2D Model | 1.09 | 86.95 |
| + Semantic Transfer | 1.31 | 88.02 |
| + Self Attention | 1.83 | 88.26 |
| + Hybrid Attention | 1.83 | **89.14** |

$T_{mixed}$ as the adapter outputs without any attention mechanism. *(4) + Self Attention:* In this phase, we adopt standard self attention with $T_{mixed}$ only. *(5) + Hybrid Attention:* We replace the attention module in the last phase with proposed hybrid attention. We can observed that neither standalone 2D nor 3D models achieve satisfactory results, while their naive combination yields significant performance improvements. Besides, our proposed multi-scale 2D semantic transfer and hybrid attention adapter effectively enhance model capabilities. More ablation studies are presented in Sec. D.

## 5. Discussion

According to our proposed modules, semantic prompts from 2D models can introduce additional semantic cues to 3D pretrained models through semantic transfer. However, an intriguing question arises: what does the 3D model actually learn from these semantic prompts? We visualize the 3D features processed by Hybrid Attention Adapter in the last layer of the proposed paradigm. As shown in Figure 4, the feature colors are transformed into feature space using PCA,

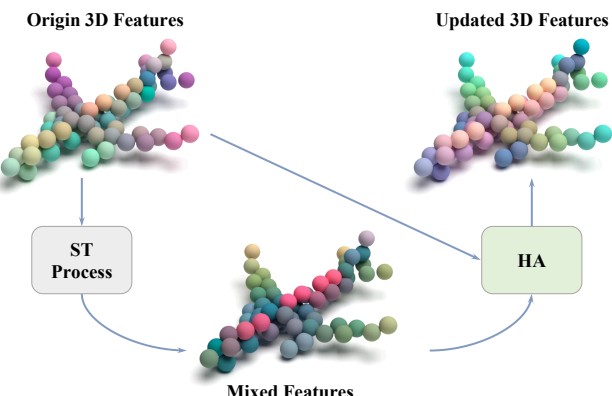

*Figure 4.* We select features from the final layer of HAA for visualization. Short ST denotes semantic transfer, and HA denotes hybrid attention.

where the same color indicates feature consistency. The visualization results show that the injected 2D semantic information effectively aligns features of identical structures, and such feature distributions significantly enhance the generalization capability of pretrained models on downstream tasks. Meanwhile, the newly proposed hybrid attention mechanism strengthens semantic associations while preserving the integrity of 3D information, further improving the effectiveness of cross-modal fusion.

## 6. Limitations

As shown in Table 8, we calculate the FLOPs of models. Due to the participation of 2D pretrained models (ViT-B/16 & ViT-B/14), our method exhibits relatively high FLOPs. To adopt off-the-shelf 2D pretrained models for saving storage and computational costs, this limitation cannot be resolved by our current design. However, our new paradigm is compatible with any combination of transformer-based 3D and 2D pretrained models, if more lightweight and higher-performing 2D pretrained models are proposed in the future, the proposed paradigm can achieve better inference speed and performance without any modifications.

## 7. Conclusions

We propose a new paradigm that, for the first time, explores the integration of 2D visual semantic cues from frozen 2D pretrained models into efficient point cloud understanding, significantly improving their perfomance on downstream tasks while maintaining parameter efficiency. Our results shown that the porposed paradigm boosts the performance of 3D pretrained models on downstream tasks while keeping minimal trainable parameters. With off-the-shelf 2D and 3D pretrained models, our paradigm outperforms all models across different training and fine-tuning stratgies on the synthetic shape classification and challenging real-world 3D object recognition. Our method is compatible with any com-

bination of transformer-based 3D and 2D pretrained models, as more powerful large foundation pretrained models emerge in the future, our approach holds limitless potential.

## Impact Statement

This paper explores the transfer of rich semantic knowledge from 2D pre-trained models to enhance the generalization capability of 3D pre-trained models on downstream tasks, under a parameter-efficient paradigm. To the best of our knowledge, this represents the first attempt in the field of parameter-efficient transfer learning for point cloud understanding. Besides, the proposed paradigm is compatible with any combination of transformer-based 3D and 2D pretrained models, as more powerful large foundation pretrained models emerge in the future, the approach holds limitless potential.

## Acknowledgments

This work was partially supported by National Defense Basic Scientific Research program(No.JCKY2022911B002)

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

## A. Additional Relatedwork

We have briefly introduced the AT paradigm in the Section 3.2. Here, we introduce another paradigm, Prompt Tuning (PT). As shown in Figure 5, PT generates a set of tokens as prompts through random initialization. These prompts are added to the input of transformer blocks or attention layers and interact with the original tokens through the self-attention mechanism. During fine-tuning, the weights of the backbone network remain frozen, and only the weights of the prompts are updated.

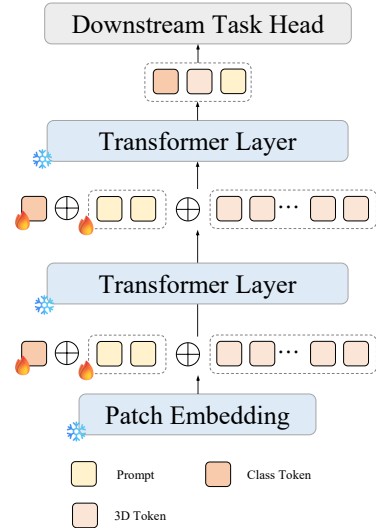

*Figure 5.* **The illustration of Prompt Tuning.**

## B. Additional Module Explanation

### B.1. Modality Alignment

Here, we will introduce the classification task for aligning modality semantics in detail. As shown in Figure 6(a), in shape classification tasks, such as downstream tasks on ScanObjectNN and ModelNet40, we concatenate 2D semantic prompts $i_{pro}$ of the last layer with origin classification head inputs $T_{input}$. Besides, in the part segmentation task, we add an additional classification head for performing semantic alignment of 2D semantic prompts. As shown in Figure 6(b), $I_{pro}$ are fed into a classification head, and the pipeline of part segmentation is not affected.

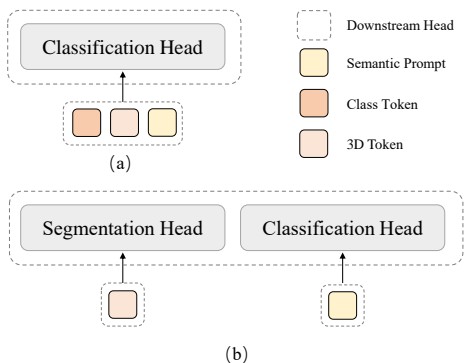

*Figure 6.* **The details of semantic alignment.**

### B.2. 3D-to-2D Projection

First, when the intrinsic and extrinsic parameters of view $v$ are determined, we can establish the camera coordinate system and the target 2D image coordinate system for projection. Given the point cloud, we can also determine the world coordinate system. The first step involves transforming the point cloud into the camera coordinate system. Let the mapped point

coordinate be $(X_w, Y_w, Z_w)$, with $R$ representing the rotation matrix and $t$ the translation vector in the extrinsic parameters matrix, we can obtain the point coordinate $(X_c, Y_c, Z_c)$ in camera coordinate system:

$$
\begin{bmatrix} X_c \\ Y_c \\ Z_c \\ 1 \end{bmatrix} = \begin{bmatrix} R_{11} & R_{12} & R_{13} & t_x \\ R_{21} & R_{22} & R_{23} & t_y \\ R_{31} & R_{32} & R_{33} & t_z \\ 0 & 0 & 0 & 1 \end{bmatrix} \cdot \begin{bmatrix} X_w \\ Y_w \\ Z_w \\ 1 \end{bmatrix} \tag{9}
$$

After obtaining the coordinates of points $(X_c, Y_c, Z_c)$ in the point cloud within the camera coordinate system, we can project them onto the 2D image plane using the camera's intrinsic matrix.

$$
\begin{bmatrix} u \cdot Z_c \\ v \cdot Z_c \\ Z_c \end{bmatrix} = \underbrace{\begin{bmatrix} f_x & s & c_x \\ 0 & f_y & c_y \\ 0 & 0 & 1 \end{bmatrix}}_{\text{Intrinsic Matrix}} \cdot \begin{bmatrix} X_c \\ Y_c \\ Z_c \end{bmatrix} \quad \Rightarrow \quad \begin{cases} u = \dfrac{f_x X_c + s Y_c}{Z_c} + c_x \\ v = \dfrac{f_y Y_c}{Z_c} + c_y \end{cases} \tag{10}
$$

The $(u, v)$ is the corresponding 2D coordinate of 3D point $(X_w, Y_w, Z_w)$.

## C. Additional Implementation Details

Table 5. **Training recipes for Parameter-Efficient Transfer Learning.**

| Config | ScanObjectNN | ModelNet40 | ModelNet40-FewShot | ShapeNetPart |
|---|---|---|---|---|
| optimizer | AdamW | AdamW | AdamW | AdamW |
| learning rate | 2e-5 | 1e-5 | 1e-5 | 2e-4 |
| weight decay | 5e-2 | 5e-2 | 5e-2 | 5e-2 |
| learning rate scheduler | cosine | cosine | cosine | cosine |
| training epochs | 300 | 300 | 150 | 300 |
| warmup epochs | 10 | 10 | 10 | 10 |
| batch size | 32 | 32 | 32 | 16 |
| drop path rate | 0.2 | 0.1 | 0.1 | 0.1 |
| Generator rank | 16 | 16 | 16 | 16 |
| q rank of HAA | 18 | 18 | 18 | 18 |
| k rank of HAA | 18 | 18 | 18 | 18 |
| BN-v rank of HAA | 64 | 72 | 32 | 128 |
| image resolution | 224×224 | 224×224 | 224×224 | 224×224 |
| image patch size | 16/14 | 16/14 | 16/14 | 16/14 |
| number of points | 2048 | 1024 | 1024 | 2048 |
| number of point patches | 128 | 64 | 64 | 128 |
| point patch size | 32 | 32 | 32 | 32 |
| augmentation | Scale&Trans/Rotation | Scale&Trans | Scale&Trans | - |
| GPU device | GTX 3090 | GTX 3090 | GTX 3090 | GTX 3090 |

We adopt downstream fine-tuning configuration following pioneer work PointMAE (Jiang et al., 2023). More details are provided in 5. Performing fine-tuning on ScanObjectNN (Uy et al., 2019) as an example, the overall training includes 300 epochs, with a cosine learning rate (Loshchilov & Hutter, 2016) of 5e-4, and a 10-epoch warm-up period. We adopt AdamW (Loshchilov, 2017) as the optimizer. Besides, we show the BN rank of our proposed Hybrid Attention Adapter (HAA) and the rank of $\alpha$ and $\beta$ generator, which is the dimension of the feature passed through downward projection. We also provide relevant setup of 3D-to-2D projection and 2D pretrained models, such as the resolution of 2D depth maps and the image patch size of 2D transformers.

# D. Additional Ablation Studies

## D.1. Ablation Study of Self Attention in Adapter

In this section, we conduct experiments on ScanObjectNN (Uy et al., 2019) to investigate the effectiveness of incorporating self-attention mechanisms within the adapter architecture without introducing 2D semantic cues. This exploration aims to determine whether the self-attention mechanism can enhance the model's generalization capability in the absence of additional semantic cues. We chose PointMAE (Pang et al., 2022) with DAPT (Zhou et al., 2024) as our 3D baseline. As shown in Table 7, adopting self attention only bring limited performance improvement, which clarifies the importance of 2D semantic cues.

*Table 6.* **Additional ablation study of the attention mechanism.**

| Method | #TP (M) | PB_T50_RS |
|---|---|---|
| 3D Baseline | 1.09 | 85.08 |
| + Self attention | 1.61 | 85.53 |

## D.2. Ablation Study of BN-v Rank

We design the BN-v follow the configuration of adapter in DAPT (Zhou et al., 2024).

*Table 7.* **Ablation study of BN-v rank.**

| Method | #TP (M) | PB_T50_RS |
|---|---|---|
| 8 | 1.32 | 87.82 |
| 16 | 1.39 | 87.98 |
| 32 | 1.54 | 88.31 |
| 64 | 1.83 | 89.14 |

## D.3. Ablation Study of FLOPs

*Table 8.* **Ablation study of FLOPs**

| Method | FlOPs | PB_T50_RS |
|---|---|---|
| PointMLP | 31.4 | 85.40 |
| PointMAE+IDPT | 7.2 | 84.94 |
| PointMAE+DAPT | 5.0 | 85.05 |
| PointMAE+CLIP(ViT-B/32) | 12.9 | 88.82 |
| PointMAE+CLIP(ViT-B/16) | 22.6 | 89.14 |

## D.4. Ablation Study of Hybrid Attention

In some cross-modal attention mechanisms (Lu et al., 2019), features from one modality are typically used as queries, while features from the other modality serve as keys and values. However, after enhancing the features through Semantic Transfer, this approach is no longer the optimal solution.

*Table 9.* **Ablation study of hybrid attentinon**

| Method | #TP (M) | PB_T50_RS |
|---|---|---|
| Cross-modal attention | 1.8 | 88.22 |
| Hybrid attention | 1.8 | **89.14** |

# E. Additional Discussion

During our investigation, we observed that the performance improvements brought by our proposed paradigm on ScanOb-jectNN's hardest split and ModelNet40 significantly surpass those achieved on the objbg and objonly splits. To analyze the underlying reasons, we visualize the 2D depth maps generated through our 3D-to-2D projection. As shown in Figure 7, the ModelNet40, as a synthetic dataset, produces exceptionally high-quality depth maps through rendering. This characteristic fully leverages the capabilities of 2D models, consequently yielding substantial performance gains in synthetic classification and segmentation tasks. Furthermore, as demonstrated in Figure 9, Figure 10 and Figure 8, the additional noise introduced in the hardest split exhibits minimal impact on imaging quality and 2D model classification performance (Table 10) after planar projection. This observation suggests that 2D models can effectively filter noise patterns that prove challenging for 3D models to process, thereby significantly enhancing model robustness in complex environments.

*Table 10.* **Additional shape classification results on ScanObjectNN and Modelnet40 with only CLIP.**

| Dataset | #TP (M) | Accuracy (%) |
|---|---|---|
| OBJ_ONLY(Scan) | 0.27 | 84.22 |
| OBJ_BG(Scan) | 0.27 | 84.48 |
| PB_T50_RS(Scan) | 0.27 | 84.13 |
| Modelnet40 | 0.27 | 92.78 |

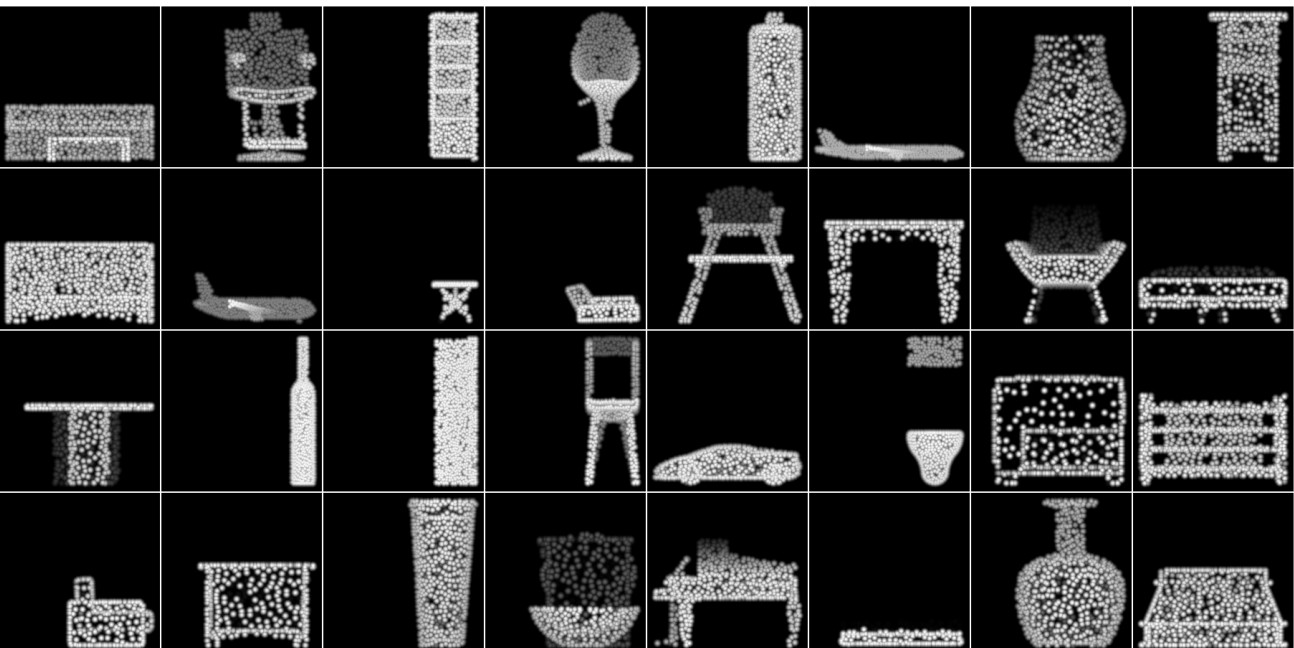

*Figure 7.* **Depth maps of Modelnet40.**

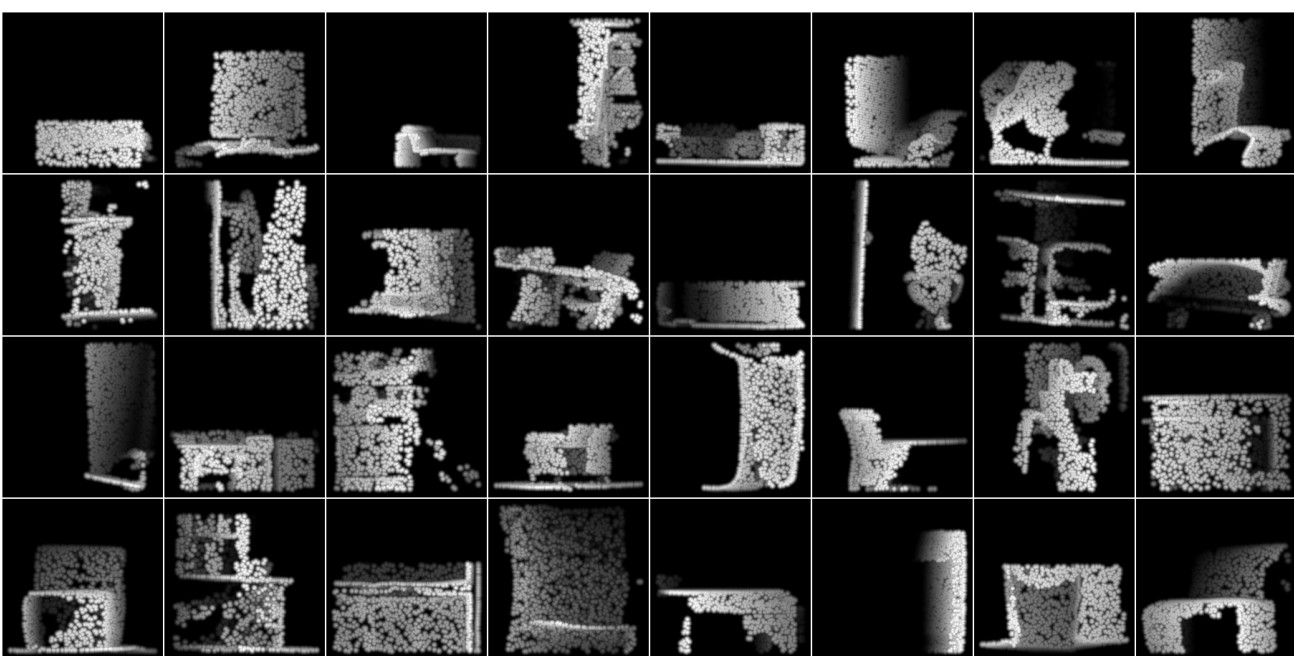

*Figure 8.* **Depth maps of ScanObjectNN-objonly.**

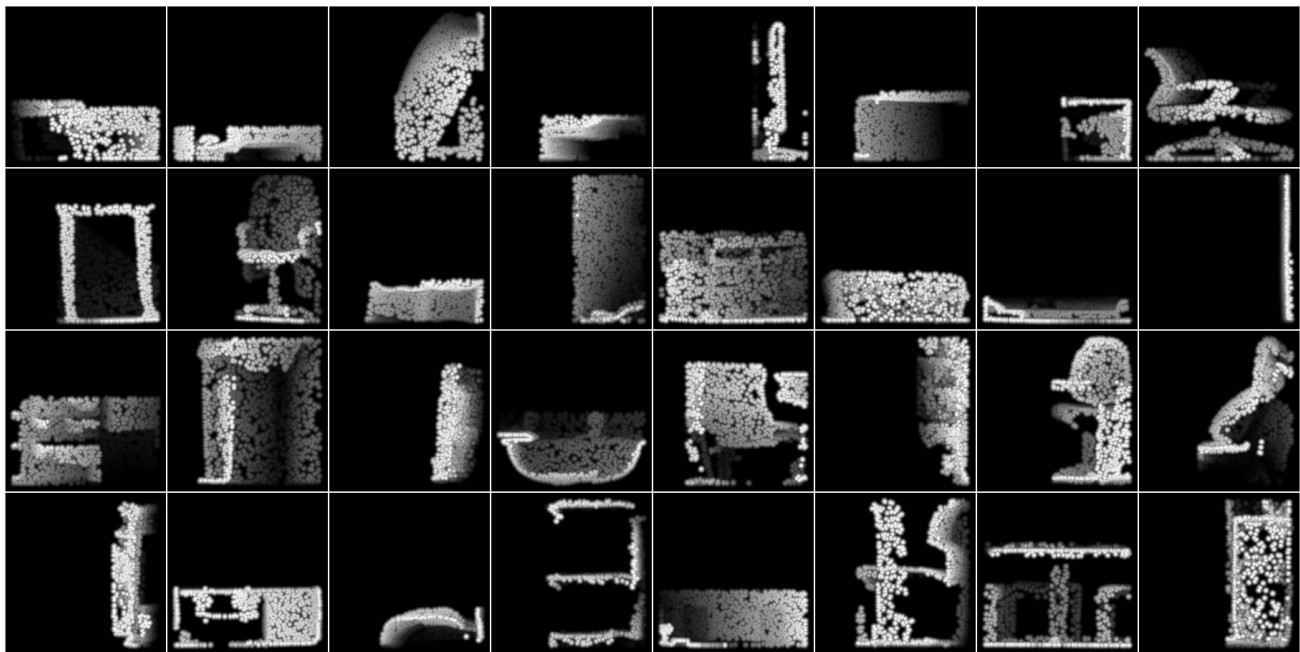

*Figure 9.* **Depth maps of ScanObjectNN-objbg.**

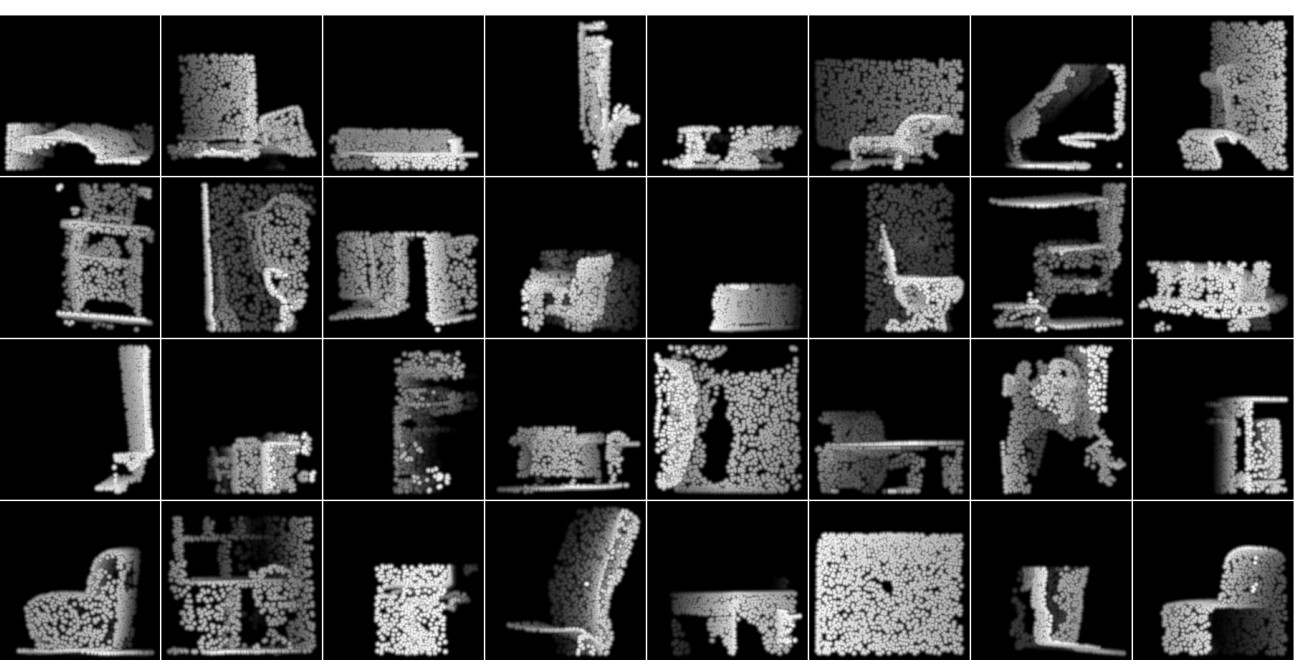

*Figure 10.* **Depth maps of ScanObjectNN-hardest.**

