# OpenReview forum: "Exploring Vision Semantic Prompt for Efficient Point Cloud Understanding"
_ICML.cc/2025/Conference — ICML 2025 poster_

### Official Review · Reviewer_8Bgm · 2025-02-26

**Overall Recommendation:** 2

**Summary:**

This paper explores PETL for 3D point cloud understanding to address the issues of full fine-tuning, such as forgetting pretrained knowledge and high storage costs. The authors propose a novel fine-tuning paradigm that integrates 2D vision semantic prompts from frozen pretrained models to improve the performance of 3D models with minimal trainable parameters.

**Claims And Evidence:**

1. The authors believe that rich semantic cues in 2D pretrained models can improve the generalization of 3D point cloud understanding. While the experimental results, including classification and part segmentation, indicate the improvements, it is not entirely clear if the improvement is consistent across different type of 3D tasks, such as 3D point cloud semantic segmentation and object detection. Moreover, the datasets used, such as ModelNet40, ScanObjectNN and ShapeNetPart, primarily contain object-centric point clouds, which may not be able to fully capture real-world scenarios.
2. The paper claims that the projection of 3D point cloud to three orthogonal depth images can mitigate global ambiguities, and the semantic transfer module can mitigate local ambiguities. However, there is no experimental results that can prove it.

**Essential References Not Discussed:**

N/A

**Experimental Designs Or Analyses:**

Strengths:
1. The authors make a fair comparison with many baselines, including supervised learning-only, full fine-tuning and PETL methods.
2. The authors conduct an ablation study for each proposed module.
Issues:
1. Need experiments on more complex real-world scenarios.
2. More complex 3D scene-level tasks to prove the effectiveness.

**Methods And Evaluation Criteria:**

1. The proposed method requires the low number of trainable parameters as shown in the experiment section, however, the efficiency of the model should be further validated by comparing the inference time with baselines.
2. To prove the effectiveness of the proposed model, it should be evaluated on more large-scale scene understanding tasks.

**Other Comments Or Suggestions:**

1. Typo: "PoinrMamba" in the table 2.
2. Type: "perfomance" in lines 368, 371, 427.

**Other Strengths And Weaknesses:**

Strengths:
1. HAA provides a new way to efficiently fuse 2D features with 3D point cloud features.
2. The proposed method outperforms the baselines on three datasets (ScanObjectNN, Modelnet40 and ShapeNetPart), and it is also efficient in terms of trainable parameters.
3. The figures and tables are clear and easy to understand.
Weaknesses:
1. No experiments or direct visualization to show how the model solves the global and local ambiguity.
2. For few-shot learning and part segmentation tasks, the improvement is limited.

**Questions For Authors:**

1. Since 2D pretrained models are trained on RGB images, how can they effectively extract features from depth images?
2. Considering that depth images lack background information and contain only a single object, how can meaningful 2D semantic features be extracted and fused to enhance 3D point cloud understanding?

**Relation To Broader Scientific Literature:**

The paper introduces the HAA to fuse 2D vision semantics into 3D point cloud features, and the trainable parameters are relatively low compared to other PETL and full fine-tuning methods.

**Theoretical Claims:**

The mathematical formulations, including all the equations, are clear to me.

---

> ### Author Rebuttal · Authors · 2025-04-01
>
> We would like to thank the reviewer for the detailed review and valuable suggestions.
>
> **Mitigation of local and global ambiguity:** We understand the reviewer's concern and appreciate the opportunity to clarify. During the classification of 100 airplane and chair samples, we recorded which view the 2D prompt selected by max pooling originated from, and counted the number of times each view was chosen.The statistics in the Table below are based on the ModelNet40 dataset.
>
> | Category | Num. of Top | Num. of Side | Num. of Front |
> | --- | --- | --- | --- |
> | Airplane | 67 | 11 | 22 |
> | Chair | 17 | 48 | 35 |
>
> We observed that when class tokens from three views are simultaneously fed into the network, the trained model tends to favor specific views for different categories—for instance, the top view for airplanes and front/side views for chairs. These preferred views typically exhibit the least local ambiguity. Besides, our approach enables the model to adaptively associate each object with its most informative view, thereby mitigating global ambiguity. We will include more detailed technical explanations in the revised version.
>
> **Comparison of inference time:** We agree with the reviewer that inference time is an important metric. Just in case, we respectfully clarify that the primary goal of "efficienct" in parameter-efficient fine-tuning is to mitigate the storage overhead caused by the exponential growth of parameters in large foundation models. To make our work more comprehensive and complete, we will add a new Table for comparison of inference time on Modelnet40 based on your suggestions.
>
> | Method | FLOPs (G) | Infer. Time (Per obj.) |
> | --- | --- | --- |
> | PointMAE + DAPT | 4.8 | 0.054s |
> |  + 3D-2D Projection + CLIP(ViT-B/16) | 22.6 | 0.081s |
> | +Semantic Transfer | 22.6 | 0.083s |
> | +Hybrid Attention | 22.6 | 0.092s |
>
> **Performances gain on segmentation and few-shot classification:** We understand the reviewer's concern regarding the few-shot and part segmentation performances. The ShapeNetPart and ModelNet40 (few-shot for PETL) are nearly saturated datasets, where multiple methods in recent years have achieved similar performance. Additionally, the ShapeNetPart contains some commonly recognized annotation errors, leading to discontinuous performance improvements.
>
> **Concerns regarding 2D pretrained models processing depth maps:** We understand the reviewer's concern regarding the difficulty for 2D models to extract semantic information from depth maps. 2D foundation models are pretrained on large-scale image datasets and possess generalizable feature representations. In our method, we convert single-channel depth maps into three-channel pseudo-RGB images by repeating the depth values across channels. Besides, as shown in Figure 7, the proposed 3D-to-2D projection strategy mitigates issues such as texture loss and boundary blurring. According to our feasibility experiments, the pretrained 2D model can effectively extract semantic cues from the depth maps we constructed. In addition, prior works such as [PointCLIP: Point Cloud Understanding by CLIP, CVPR2022]  and [Learning 3D Representations from 2D Pre-trained Models via Image-to-Point Masked Autoencoders, CVPR2023]have explored the use of CLIP to extract semantic information from depth maps generated by projecting point clouds.
>
> | 2D Model | ModelNet40 (%) |
> | --- | --- |
> | CLIP | 92.78 |
> | DINOv2 | 91.86 |
>
> **Effectiveness of 2D semantics for point cloud understanding:** We acknowledge the reviewer's concern about whether 2D semantics can meaningfully enhance 3D point cloud representation. Currently, Transformer-based point cloud foundation models are pretrained on ShapeNet, a dataset composed of background-free, single-object point clouds. Besides, the depth maps used in our method are generated directly from the 3D point clouds through a globally consistent projection, ensuring a correspondence between pixels and 3D points.
>
> **Real-world applications:** We understand the reviewer's concerns regarding our application scenario and appreciate the opportunity to provide further clarification. We agree that scene-level applications are an important future direction. Existing Transformer-based methods are mainly pretrained on ShapeNet, while large-scale scene-level models remain limited. For a fair comparison with previous PEFT methods, we validate our method on the same datasets. Our proposed method has promising applications in various tasks. For example, in robotic interaction, the first step involves cropping the target object followed by classification without background, and it is necessary to understand the semantic parts of the object for interaction.
>
> **We hope our responses have addressed your concerns and helped clarify any confusion. We truly welcome any additional questions or suggestions that could further improve our work.**

---

### Official Review · Reviewer_5QCm · 2025-03-09

**Overall Recommendation:** 4

**Summary:**

This paper proposes to leverage the rich semantic cue inherent in pretrained 2D foundation models as the semantic prompt for efficient point cloud learning. The authors identify the limitation of existing 3DPEFT research: they fail to align semantic relationships of features required by downstream tasks. To address this issue, a new Hybrid Attention Adapter is designed to fuse 2D semantic cues into 3D representations efficiently with minimal trainable parameters. Experiment results demonstrate the effectiveness of the proposed hybrid attention.

**Claims And Evidence:**

The authors claim that the current full-tuning method will forget the pretrained knowledge,  while the existing PEFT methods fall short in aligning semantic relationships. However, these claims are not supported by convincing evidence or theoretical analysis. Why would the feature inconsistency affect the downstream performance in Figure 1?

**Essential References Not Discussed:**

No

**Experimental Designs Or Analyses:**

Yes, the experimental design is sound and comprehensive.

**Methods And Evaluation Criteria:**

The design of the framework and evaluation make sense.

**Other Comments Or Suggestions:**

1. In the session of  3D-to-2D Projection, the authors employ a straightforward strategy to fuse all three orthogonal depth maps. Considering the ambiguity as illustrated in Figure 1(c), is the choice of three views already satisfied in any downstream tasks? I wonder what will happen if we adjust the view direction or the number of views, such as down-to-top or front-to-back views.
2. The calculation of the number of tunable parameters in Table 1 may introduce confusion. While the number of trainable parameters increases, the relative percentage decreases.
3. The ablation in Table 4 is kind of unintuitive; the baseline should be the 3D model only instead of using the 2D model.

**Other Strengths And Weaknesses:**

Despite the intuitive and practical idea of using 2D semantic prior to enhance 3D representation learning, this paper also conducted extensive evaluation to validate its design. I recommend an accept for this paper, for which I believe its further impact in this topic.

**Questions For Authors:**

See the comments above.

**Relation To Broader Scientific Literature:**

This paper introduces a promising way to incorporate 2D semantic prior into 3D representation learning. While cross-modal complementary information has been widely explored in a wide range of research areas, it has rarely been investigated in the field of 3D parameter-efficient learning.

**Theoretical Claims:**

There is no theoretical analysis in this paper.

---

> ### Author Rebuttal · Authors · 2025-04-01
>
> We greatly appreciate the reviewer's thoughtful feedback and sincerely thank the reviewer for the positive recognition of our work.
>
> **Feature inconsistency affects downstream task performance**：We appreciate the reviewer's question regarding the impact of feature inconsistency on downstream performance. There exists a mismatch between the representations emphasized by the pre-training task and those required by the downstream task. Pre-training often focuses on local structures and exhibits a strong positional correlation. For example, the wing features of an airplane are different before fine-tuning due to they are not adjacent in 3D space, even though their structures are similar. However, downstream tasks rely more on semantic features, which require modeling long-range dependencies — capturing the relationship between the two wings of an airplane.
>
> **Ablation on number of views and impact of viewpoints:** We greatly appreciate the reviewer's attention to the impact of view count—this is indeed a very important aspect for analysis. We additionally conduct experiments using only the top view, six views, and nine views. The results show that increasing the number of views does not lead to performance improvement. In feasibility experiments, we observed that orthogonal views yield the best 2D classification results. It provides a balanced and comprehensive representation of the 3D structure while maintaining geometric fidelity. Therefore, redundant views are filtered out by max pooling and do not contribute to the final performance.
>
> | Number of views | FLOPs (G) | PB_T50_RS (%) | ModelNet40 (%) |
> | --- | --- | --- | --- |
> | 1 (Top view) | 22.6 | 88.16 | 94.5 |
> | 3 | 22.6 | 89.14 | 95.2 |
> | 6 | 22.6 | 88.97 | 95.2 |
> | 9 | 22.6 | 89.18 | 94.9 |
>
> Regarding the question on different viewing directions, in datasets such as ModelNet40, objects are aligned orthogonally but their orientations are not consistent — there can be angular differences like 90 or 180 degrees among objects within the same category. As a result, the choice of viewing direction has minimal impact on the final performance. Additionally, we explored projecting 3D point clouds onto cylindrical and spherical surfaces. However, these projections introduced significant distortions, which degraded the quality of the rendered views and adversely affected 2D models' performances.
>
> **Confusing description in Table 1:** We acknowledge that the calculation of the percentage of tunable parameters in Table 1 has caused confusion. The percentage listed for "Ours" refers to the model after incorporating the 2D model. We will provide a more detailed explanation in future versions to clarify this point. Thank you for your helpful suggestion!
>
> **Unintuitive ablation table:** We agree that the original ablation may not be sufficiently intuitive. Therefore, we have revised the ablation study of the main modules and present the updated results in the following table:
>
> | Method | #TP(M) | PB_T50_RS (%) |
> | --- | --- | --- |
> | 3D Baseline | 1.07 | 85.18 |
> | + 2D Baseline | 1.09 | 86.95 |
> | + Semantic Transfer | 1.31 | 88.02 |
> | + Self Attention | 1.83 | 88.26 |
> | + Hybrid Attention | 1.83 | 89.14 |
>
> **We sincerely thank the reviewer for the positive recognition of our work. We truly value your comments and would appreciate any further questions or suggestions that can help us improve the paper.**

---

### Official Review · Reviewer_5f8a · 2025-03-10

**Overall Recommendation:** 3

**Summary:**

This paper introduces a parameter-sparse fine-tuning strategy for fine-tuning pre-trained 3D point cloud models with vision semantic prompts of a frozen vision transformer in the 2D domain. It introduces a new Hybrid Attention Adapter (HAA) which aggregates the features of different views of the depth images in order to achieve state-of-the-art performance on ScanObjectNN and ModelNet40 with minimal trainable parameters (~1–2%).

**Claims And Evidence:**

The arguments are substantiated by rigorous experiments with a variety of tasks (classification, segmentation, few-shot learning). The robust empirical results clearly demonstrate the strength of combining 2D semantic features with 3D models.

**Essential References Not Discussed:**

The paper failed to mention "PointCLIP (Zhang et al., CVPR 2022)," an important prior work also employing CLIP-based 2D prompts for the sake of 3D tasks. Citing it will serve to convey how this paper generalizes over the previous prompting approaches.

**Experimental Designs Or Analyses:**

The experimental configurations are extensive and solid, with broad comparison with the most up-to-date strategies and multiple backbones. Ablation tests clearly demonstrate the incremental advantage of each component proposed.

**Methods And Evaluation Criteria:**

The proposed approaches (multi-view depth projection, vision semantic prompting, HAA) are well-suited and well-organized for addressing the problem of semantic alignment in parameter-efficient tuning of the 3D model. The typical benchmarks used (ScanObjectNN, ModelNet40, ShapeNetPart) are well-suited and well-known in the field.

**Other Comments Or Suggestions:**

Minor typos:
"deitals" -> "details", in page 4.
"Vsion" -> "Vision", in the title on OpenReview, not in the paper.
Directions for future work: explore usage at the scene level, leverage RGB/color data, or use language prompts for richer semantics.

**Other Strengths And Weaknesses:**

Strength
Strong integration of multi-modal semantic prompting with PETL.
Current results for top benchmarks.
Careful experimental validation with judicious ablations.
Well written with great supporting material.

Weakness
No code released yet, limiting short-term replicability.
Computational complexity for inference due to the utilization of multiple views and ViT.
Constrained gains for the segmentation tasks compared to classification tasks.
Practical application can be thwarted by the intricateness of the architecture.

**Questions For Authors:**

1. Have you taken into account the computational cost of your strategy in the context of multiple views as well as the ViT? How realistic is this for real-world application?
2. Did you use fewer or more than three views? How do results vary with the number of views?
3. Qualitatively describe in detail the specific semantic cues the 2D prompts provide which significantly facilitate the 3D recognition tasks.
4. CLIP generally outperformed DINOv2 prompts. Do we credit this to CLIP’s multi-modal pre-training? What do you think of this difference?
5. How does performance scale with larger 3D models—does the 2D prompt remain complementary regardless of the size of the 3D model?

**Relation To Broader Scientific Literature:**

This work builds effectively on the most recent in 3D pre-training (PointMAE, PointBERT), PETL (IDPT, Point-PEFT, DAPT), and cross-modal learning (ACT, multi-view networks). It bridges in a new manner 2D semantic prompting with parameter-efficient tuning, pushing the state-of-the-art in the combined field.

**Theoretical Claims:**

There are no significant theory proofs or hypotheses requiring formal verification. The proposed adapter and attention mechanisms are solidly grounded, clearly described, and in line with traditional architectures.

---

> ### Author Rebuttal · Authors · 2025-04-01
>
> We sincerely thank the reviewer's for the constructive comments and valuable suggestions.
>
> **Performance gain on segmentation:** We understand the reviewer's concern and appreciate the opportunity to clarify. The ShapeNetPart is a nearly saturated dataset, where multiple methods have achieved similar performance in recent years. Additionally, the dataset contains some commonly recognized annotation errors, leading to discontinuous performance improvements. However, our method achieves better-fine-tuned performance than baseline (full fine-tuning) on the datasets with less trainable parameters.
>
> **Computational cost and practical applications:** Thank you for raising this important point regarding the computational cost of our approach; we present the FLOPs of selected models in Table 8, and we will add a new Table for comparison of inference time on Modelnet40 in the future version. Besides, we will enrich Table 1 with the comparison of FLOPS.
>
> | Method | FLOPs (G) | Infer. Time (Per obj.) |
> | --- | --- | --- |
> | PointMAE + DAPT | 4.8 | 0.054s |
> |  + 3D-2D Projection + CLIP(ViT-B/16) | 22.6 | 0.081s |
> | +Semantic Transfer | 22.6 | 0.083s |
> | +Hybrid Attention | 22.6 | 0.092s |
>
> Our proposed method has promising applications in various tasks. For example, in robotic interaction, typically, the first step involves cropping the target object, followed by classification. For interaction, it is necessary to understand the semantic parts of the object, i.e., part segmentation. The 2D pretrained models used in our work have been widely deployed in real-world scenarios, including Google's multimodal robotic system PALM-E [PALM-E: An Embodied Multimodal Language Model].
>
> **Ablation on the number of views:** We greatly appreciate the reviewer's attention to the influence of view count—this is indeed an important aspect. We additionally conduct experiments using only the top view, six views, and nine views. The results show that increasing the number of views does not lead to performance improvement. In feasibility experiments, we observed that orthogonal views yield the best 2D classification results. Therefore, redundant views are filtered out by max pooling and do not contribute to the final performance. Using only a single view may be suboptimal, as the most informative viewpoint varies across object categories. For example, the top view provides the richest semantic information for airplanes, while the side or front view may be more informative for chairs. Therefore, providing three orthogonal views and allowing the network to adaptively select the most relevant one offers a more robust and effective solution.
>
> | Number of views | FLOPs (G) | PB_T50_RS (%) | ModelNet40 (%) |
> | --- | --- | --- | --- |
> | 1 (Top view) | 22.6 | 88.16 | 94.5 |
> | 3 | 22.6 | 89.14 | 95.2 |
> | 6 | 22.6 | 88.97 | 95.2 |
> | 9 | 22.6 | 89.18 | 94.9 |
>
> During the classification of 100 airplane and chair samples, we recorded which view the 2D prompt selected by max pooling originated from, and counted the number of times each view was chosen. The statistics in the Table below are based on the ModelNet40 dataset.
>
> | Category | Top Veiw Count | Side View Count | Front View Count |
> | --- | --- | --- | --- |
> | Airplane | 67 | 11 | 22 |
> | Chair | 17 | 48 | 35 |
>
> **Qualitative Description of the Effectiveness of 2D Semantic Prompt:** Thank you for pointing out the need for a qualitative description. In the Discussion section of our paper, we visualize the features in our proposed HAA through PCA. As shown in Figure 4, with Semantic Transfer, we can transfer the 2D semantic cues to 3D features, enriching the association of the same components (wings of the airplane) and improving model performance in downstream tasks.
>
> **CLIP outperforms DINOv2:** We agree with your point. In our feasibility experiments, frozen CLIP outperforms DINOv2 when classifying using depth maps obtained via our proposed projection. We attribute this to CLIP's strong zero-shot classification capability, and we believe this stems from its image-text alignment training.
>
> | 2D Model | ModelNet40 (%) |
> | --- | --- |
> | CLIP | 92.78 |
> | DINOv2 | 91.86 |
>
> **Larger 3D foundation model:** We utilize CLIP-L  to enhance PointGPT-L [PointGPT: Auto-regressively Generative Pre-training from Point Clouds, NIPS2023] 3D model, and still observed consistent performance improvements on ModelNet40.
>
> | Method | ModelNet40 (%) |
> | --- | --- |
> | PointGPT-L | 94.1 |
> | +DAPT | 94.2 |
> | +CLIP-L | 95.5 |
>
> **Citing PointCLIP：** PointCLIP is one of the representative works in point cloud understanding that explores the integration of 2D semantic information into 3D point cloud tasks. We appreciate the reviewer's suggestion and will include a discussion of PointCLIP in the Related Work section.
>
> **We hope our clarifications have been helpful, and we would be glad to address any further questions you may have.**

---

> > ### Comment · Reviewer_5f8a · 2025-04-02
> >
> > Thank you for the extensive explanations and additional experiments provided in your answer. The computational cost explanation and practical feasibility explanation are appreciated, and the additional ablation once again adds robustness to your method. However, the slight performance gains in segmentation tasks and constant inference complexity are areas of concern.
> >
> > Having these factors, along with originality and significance, in mind, I keep my original proposal of weak accept.

---

> > > ### Author Response · Authors · 2025-04-03
> > >
> > > We appreciate the reviewer’s acknowledgment of our clarifications and additional experiments, as well as your recognition of the originality and significance of our work. Thank you again for your valuable feedback, and best wishes for your research and future projects.

---

### Official Review · Reviewer_aPXL · 2025-03-12

**Overall Recommendation:** 3

**Summary:**

This paper introduces a novel fine-tuning paradigm for 3D pretrained models by leveraging 2D semantic prompts from frozen 2D pretrained models to enhance point cloud understanding, while maximally preserving the pretrained knowledge. The proposed method has three main contributions:

1.	Introduction of the Hybrid Attention Adapter (HAA) that effectively utilizes 2D semantic cues from pretrained 2D foundation models to guide feature aggregation in 3D models.

2.	Incorporation of depth maps to address ambiguity issues arising from viewpoint projections.

3.	Comprehensive experimental validation across multiple datasets (ScanObjectNN, ModelNet40, and ShapeNetPart), demonstrating performance improvements over existing methods in both classification and segmentation tasks.

**Claims And Evidence:**

The paper claims to address inherent local ambiguities in 3D-to-2D projection and illustrates this challenge with examples where points that are distant in 3D space appear deceptively proximate in 2D projections. This highlights a valid concern in the field. Additionally, the authors discuss knowledge forgetting in downstream tasks through PCA visualization, which I find unconvincing. They try to demonstrate information loss in semantic and geometric space through distances in PCA-reduced features. A more rigorous analysis would be necessary.

**Essential References Not Discussed:**

The proposed method relates to multi-modal fusion in 3D vision, with its key distinction being the leverage of foundation models under a fine-tuning paradigm. However, I suggest including comparisons with classical multi-modal fusion methods such as [PointPainting: Sequential Fusion for 3D Object Detection, CVPR2020] and [EPNet: Enhancing Point Features with Image Semantics for 3D Object Detection, ECCV 2020].

**Experimental Designs Or Analyses:**

The paper conducts experiments on ScanObjectNN, ModelNet40, and ShapeNetPart, demonstrating improvements over the Point-BERT and Point-MAE baselines, and highlighting the effectiveness of the proposed method. However, several weaknesses should be addressed:

1.	Marginal improvements over SOTA: Compared to state-of-the-art supervised learning-only and full-finetune methods, the performance gains are minimal. For instance, ADS achieves an accuracy of 95.1% on ModelNet40, while the proposed method achieves 95.2%. Similar marginal improvements are observed on ScanObjectNN. This raises concern about the practical significance of the proposed approach.

2.	Limited generalizability of pretrained 3D models: The paper claims to preserve general knowledge in pretrained 3D models, but these models are trained on relatively small datasets and each sample contains only a single object with limited information, such as ModelNet40 (12,311 samples) and ScanObjectNN (15,000 samples). It may not be sufficient for the model to learn truly generalizable knowledge. As discussed earlier, supervised learning-only methods can also achieve SOTA performance. To validate the claims, experiments on more diverse and complex datasets and tasks, such as indoor and outdoor 3D detection and segmentation, are necessary.

3.	Lack of computational analysis: The paper does not provide a detailed comparison of parameters and FLOPs between the 3D branch and the 2D branch that utilizes a heavy Vision Transformer (ViT-B) model. Additionally, the inference time before and after integrating the 2D branch should be reported.

**Methods And Evaluation Criteria:**

The authors propose two designs:

1.	Incorporating depth maps in the 2D branch, which effectively mitigates ambiguities in 2D projections of 3D data.

2.	Introducing the Hybrid Attention Adapter (HAA), which renormalizes 3D features using guidance from 2D semantic cues.

However, I remain skeptical about this approach, as simply performing cross-attention between 3D tokens and 2D semantic prompts might achieve comparable results with a more straightforward and less complex implementation. As evidenced in Table 9, the Hybrid-Attention mechanism shows only a marginal 0.92% improvement compared to the trivial multi-modal cross-attention approach (widely explored in 3D detection area), which represents the true incremental gain of the proposed method. This minimal improvement raises the concern about the effectiveness and necessity of the HAA component.

**Other Comments Or Suggestions:**

Some Typos Mistakes

1.	L134 Add a “.” before "However".

2.	L197 “follow” should be “following”.

3.	L428 “Our results shown” should be “Our results show”.

4.	L430 “porposed” shold be “proposed”.

5.	L434 “stratgies” should be “strategies”.

**Other Strengths And Weaknesses:**

Strengths:

1.	The paper proposes HHA, a novel approach to enhance 3D pretrained models through feature modulation like [Arbitrary Style Transfer in Real-time with Adaptive Instance Normalization, ICCV 2017]. This approach offers a new perspective on incorporating multi-modal knowledge into 3D vision.

**Questions For Authors:**

The authors should conduct experiments on larger datasets such as ShapeNet or large-scale outdoor datasets like Waymo and nuScenes. Both pretraining and fine-tuning experiments on these larger-scale datasets would help demonstrate the method's scalability and verify whether the performance improvements remain consistent across more diverse and challenging scenarios.

**Relation To Broader Scientific Literature:**

This work provides insights into two broader areas: (1) foundation models and fine-tuning strategies, particularly in leveraging foundation models from another modality to guide the fine-tuning process, and (2) multi-modal fusion in 3D vision, where the proposed Hybrid Attention Adapter (HAA) facilitates semantic information fusion through Semantic Transfer and hybrid attention mechanisms.

**Theoretical Claims:**

There are no theoretical proofs in this paper.

---

> ### Author Rebuttal · Authors · 2025-04-01
>
> Thanks for the reviewer's careful reading and detailed comments. First, we respectfully clarify that all the pretrained models in our paper are trained on ShapeNet, these works aim to explore effective approaches for building 3D foundation models. ModelNet40, ScanObjectNN, and ShapeNetPart are used only in downstream tasks for fine-tuning. We will add this clarification to the Implementation Details section in the revised version.
>
> **Clarification on PCA and knowledge forgetting:** We understand the reviewer's concern and appreciate the opportunity to clarify. PCA is used to illustrate the difference between pre-trained features and downstream task features, and it is unsuitable to explain knowledge forgetting. The goal of pretraining is to learn generalizable point cloud representations, while downstream tasks focus on task-specific features. Full fine-tuning adjusts all parameters to fit the downstream objective, which may lead the network to lose the general features acquired during pretraining. In contrast, parameter-efficient transfer learning (PETL) updates only a small subset of parameters while keeping most of the pretrained weights frozen. This allows the model to adapt to new tasks while preserving the knowledge learned during pretraining. [Overcoming catastrophic forgetting in neural networks, PNAS2017].
>
> **Confusing description of Table 9:** We acknowledge that our explanation of Table 9 may have caused confusion. In future versions, we will provide a more detailed description of the cross-attention mechanism in Table 9, where T$_{mixed}$  serve as V and K, and T as Q. In Table below, we present experiments directly applying cross-attention between 2D prompts and 3D tokens. In PEFT, simply using cross-attention does not effectively fuse 2D information with 3D features. To address this, we design a simple ST module and leverage hybrid attention for effective feature fusion.
>
> | Method | FLOPs (G) | PB_T50_RS (%) |
> | --- | --- | --- |
> | 2D Prompt-3D Token Cross Attention (2-3 CA) | 1.46 | 87.57 |
> | Semantic Transfer (ST) | 1.31 | 88.02 |
> | 2-3 CA + Hybrid Attention(HA) | 1.98 | 88.26 |
> | ST + HA | 1.83 | 89.14 |
>
> **Marginal improvements over SOTA:** Our method proposes a parameter-efficient transfer learning framework that can be applied to existing self-supervised representation learning methods. We respectfully note that a direct comparison with fully supervised methods or self-supervised point cloud representation learning methods may not be entirely fair, as those methods often involve significantly more tunable parameters and different architectural designs. Compared to the previous SOTA method (DAPT), our method achieves higher performance on the same pretrained models. Additionally, unlike full fine-tuning, our approach requires significantly fewer parameters for downstream adaptation.
>
> **Lack of computational analysis:**  We also consider the analysis of computational cost to be highly important. We have presented FLOPs of selected models in Table 8, and we will add a new Table for comparison of inference time on Modelnet40 based on the reviewer's suggestions. Besides, we will enrich Table 1 with the comparison of FLOPs.
>
> | Method | FLOPs (G) | Infer. Time (Per obj.) |
> | --- | --- | --- |
> | PointMAE + DAPT | 4.8 | 0.054s |
> |  + 3D-2D Projection + CLIP(ViT-B/16) | 22.6 | 0.081s |
> | +Semantic Transfer | 22.6 | 0.083s |
> | +Hybrid Attention | 22.6 | 0.092s |
>
> **Scene-level applications:** We agree that scene-level applications are an important future direction. However, existing Transformer-based methods are mainly pre-trained on ShapeNet, while large-scale scene-level pretraining methods remain limited. For a fair comparison with previous PEFT methods, we validate our method on the same datasets. We also look forward to the development of Transformer-based scene-level pretraining methods. Besides, our proposed method has promising applications in real-world applications. For example, in robotic interaction, the first step involves cropping the target objects, followed by classification without background. Then, for interaction, it is necessary to understand the semantic parts of the object, i.e., part segmentation. The 2D pretrained models used in our work have been widely deployed in real-world scenarios, including Google's multi-modal robotic system PALM-E [PALM-E: An Embodied Multi-modal Language Model].
>
> **Comparisons with classical multi-modal fusion methods:** PointPainting and EPNet are classical multi-modal fusion methods. They utilize 2D semantic information to enhance scene-level 3D object detection tasks. In contrast, our work focuses on parameter-efficient transfer learning for 3D foundation models. We will include a discussion in the Method section to highlight the key differences between our approach and these methods.
>
> **We hope our replies have resolved the reviewer's concerns and provided sufficient clarification. We sincerely welcome any further questions or feedback.**

---

### Decision · Program_Chairs · 2025-05-01

**Decision:**

Accept (poster)

**Comment:**

This paper presents a novel and efficient fine-tuning paradigm for 3D pretrained models by leveraging 2D semantic prompts from frozen vision transformers, introducing a Hybrid Attention Adapter (HAA) to enhance semantic alignment with minimal trainable parameters. The method is well-motivated, experimentally validated across multiple benchmarks, and contributes meaningfully to the growing literature on cross-modal foundation model adaptation. While the performance gains are sometimes marginal, the approach is conceptually solid and opens promising directions. To strengthen the paper, the authors are encouraged to include runtime and FLOPs analysis,  ablations on the number of depth views, and evaluations on larger, scene-level datasets.